# Lyapunov-based Safe Policy Optimization

## Abstract

In many reinforcement learning applications, it is crucial that the agent interacts with the environment only through *safe* policies, i.e., policies that do not take the agent to certain undesirable situations. These problems are often formulated as a *constrained* Markov decision process (CMDP) in which the agent's goal is to optimize its main objective while not violating a number of safety constraints. In this paper, we propose safe policy optimization algorithms that are based on the Lyapunov approach to CMDPs, an approach that has well-established theoretical guarantees in control engineering. We first show how to generate a set of state-dependent Lyapunov constraints from the original CMDP safety constraints. We then propose safe policy gradient (PG) algorithms that train a neural network policy using deep deterministic policy gradient (DDPG) or proximal policy optimization (PPO), while guaranteeing near-constraint satisfaction at every policy update by projecting either the policy parameter or the action onto the set of feasible solutions induced by the linearized Lyapunov constraints. Unlike the existing (safe) constrained policy gradient algorithms, ours are more data efficient as they are able to utilize both on-policy and off-policy data. Furthermore, the action-projection version of our algorithms often leads to less conservative policy updates and allows for natural integration into an end-to-end PG training pipeline. We evaluate our algorithms and compare them with constrained policy optimization (CPO) and the Lagrangian method on several high-dimensional continuous state and action simulated robot locomotion tasks, in which the agent must satisfy certain safety constraints while minimizing its expected cumulative cost.

## 1 Introduction

In many real-world reinforcement learning (RL) problems, the agent must satisfy a number of constraints while minimizing its expected cumulative cost. In particular, these constraints might be related to *safety*, i.e., the policy learned by the agent should not take it to certain undesirable parts of the state and/or action space. In some problems, it is not only important that the policy learned by the agent (that will be deployed) be safe (Amodei et al., 2016), but also crucial that all the policies generated during training (with which the agent interacts with the environment to collect samples) be safe (Achiam et al., 2017). For example, a robot should always avoid taking actions that irrevocably harm its hardware. We often formulate the agent's interaction with the environment in constrained sequential decision-making problems as a constrained Markov decision process (CMDP). CMDPs are extensions of MDPs in which, in addition to the original cost function, there exists a constraint cost function, whose expected cumulative value should remain bounded. The additional constraint cost function gives more flexibility to CMDPs in modeling problems with *trajectory-based* constraints compared to approaches that artificially re-cast the cost of MDPs to enforce the constraints (Regan & Boutilier, 2009). Under the CMDP framework, we consider a policy to be *safe* if it satisfies the (expected) cumulative cost constraints.

A common approach to solve CMDPs is to use the Lagrangian method (Altman, 1998; Geibel & Wysotzki, 2005) that augments the original objective function with a penalty on constraint violation and computes the saddle-point of the constrained policy optimization via primal-dual method (Chow et al., 2017). Although safety is ensured when the policy converges *asymptotically*, a major drawback of this approach is that it makes no guarantee with regards to the safety of the policies generated during training. To address this issue (safety during training), heuristic algorithms for policy search with safety constraints have been proposed (e.g., Uchibe & Doya 2007). While some of them work reasonably well in practice, none is theoretically grounded to generate constraint-satisfying policies. Achiam et al. (2017) recently proposed the *constrained policy optimization* (CPO) method that extends the trust-region policy optimization (TRPO) algorithm (Schulman et al., 2015a) to handle CMDP constraints. Although there is no theoretical guarantee that the policies generated by CPO are all safe, the empirical results reported in the paper are promising in terms scalability, performance,

and constraint satisfaction, both during training and after convergence. Moreover, CPO is based on a principled approach to generate constraint-satisfying policies, and not just on heuristics. However, the CPO methodology is closely connected to TRPO. While adopting this methodology to PPO for constrained optimization is straight-forward (this is exactly how we derive our SPPO algorithm in this paper), it is not clear how it can be combined with algorithms that do not belong to the family of proximal PG algorithms (PG algorithms that are regularized with relative entropy), such as DDPG.

Another recent approach to solve CMDPs, while remaining safe during training, is by Chow et al. (2018). This work is based on the notion of Lyapunov function that has a long history in control theory to analyze the stability of dynamical systems (Khalil, 1996; Neely, 2010). Using Lyapunov functions in RL was first studied by Perkins & Barto (2002), where they were used to guarantee closed-loop stability of an agent. In a recent work, Berkenkamp et al. (2017) used Lyapunov functions to guarantee that a model-based RL agent can be brought back to a "region of attraction" during exploration. Using the theoretical underpinnings of the Lyapunov approach, Chow et al. (2018) proposed two dynamic programming (DP) algorithms to solve a CMDP and provided theoretical analyses for the feasibility and performance of the resulted policies. The proposed algorithms correspond to the two celebrated DP algorithms: policy and value iteration. They then extended their algorithms to learning (the scenario in which the MDP model is unknown) and proposed RL algorithms that correspond to approximate policy and value iteration. However, since their algorithms are all value-function-based, applying them to continuous action problems is not straightforward.

In this paper, we extend the Lyapunov-based approach to solving CMDPs of Chow et al. (2018) to continuous action problems that play an important role in control and robotics. Our contributions can be summarized as follows: **1)** We formulate the problem of safe RL as a CMDP and propose a *Lyapunov-function-based* policy optimization framework that can handle continuous action CMDPs. **2)** By leveraging the theoretical underpinnings of the Lyapunov-based approach to CMDPs in Chow et al. (2018), we present two classes of safe policy optimization algorithms that can work with any standard policy gradient algorithm such as deep deterministic policy gradient (DDPG) (Lillicrap et al., 2015) and proximal policy optimization (PPO) (Schulman et al., 2017). The first class of our algorithms is based on constrained optimization w.r.t. policy parameter (similar to what is done in constrained policy optimization (CPO)). The second class hinges on the concept of a safety layer introduced by Dalal et al. (2018) and transforms the constrained policy optimization problem into an unconstrained one, by integrating the Lyapunov constraints into the policy network via safety-layer augmentation. **3)** We evaluate our algorithms and compare them to two baselines, CPO (Achiam et al., 2017) and the Lagrangian method, on several robot locomotion tasks, in which the agent must satisfy certain safety constraints while minimizing its expected cumulative cost. Our results show that our algorithms outperform the baselines in terms of balancing the performance and constraint satisfaction at every policy update.

## 2 PRELIMINARIES

We consider the RL problem in which the agent's interaction with the environment is modeled as a Markov decision process (MDP). A MDP is a tuple $(\mathcal{X}, \mathcal{A}, \gamma, c, P, x_0)$, where $\mathcal{X}$ is the state space; $\mathcal{A}$ is the action space; $\gamma \in [0, 1)$ is the discounting factor; $c(x, a) \in [0, C_{\max}]$ is the immediate cost function (negative reward); $P(\cdot|x, a)$ is the transition probability distribution; and $x_0 \in \mathcal{X}$ is the initial state. Our results easily generalize to random initial states and random costs, but for simplicity we will focus on the case of deterministic initial state and immediate cost. In a more general setting where cumulative constraints are taken into account, we define a constrained Markov decision process (CMDP), which extends the MDP model by introducing additional costs and associated constraints. A CMDP is defined by $(\mathcal{X}, \mathcal{A}, \gamma, c, d, P, x_0, d_0)$, where the components $\mathcal{X}, \mathcal{A}, \gamma, c, P, x_0$ are the same as in the unconstrained MDP; $d(x) \in [0, D_{\max}]$ is the immediate constraint cost; and $d_0 \in \mathbb{R}_{\geq 0}$ is an upper-bound on the expected cumulative (through time) constraint cost. To formalize the optimization problem associated with CMDPs, let $\Delta$ be the set of Markov stationary policies, i.e., $\Delta(x) = \{\pi(\cdot|x) : \mathcal{X} \to \mathbb{R}_{\geq 0s} : \sum_a \pi(a|x) = 1\}$ for any state $x \in \mathcal{X}$. For notational convenience, at each state $x \in \mathcal{X}$, we define the generic Bellman operator w.r.t. policy $\pi \in \Delta$ and generic cost function $h$ as $T_{\pi,h}[V](x) = \sum_a \pi(a|x)\Big[h(x, a) + \gamma \sum_{x' \in \mathcal{X}} P(x'|x, a)V(x')\Big]$.

Given a policy $\pi \in \Delta$ and an initial state $x_0$, the expected cumulative cost is defined as $\mathcal{C}_\pi(x_0) := \mathbb{E}\big[\sum_{t=0}^{\infty} \gamma^t c(x_t, a_t) \mid x_0, \pi\big]$, and the safety constraint is defined as $\mathcal{D}_\pi(x_0) \leq d_0$, where $\mathcal{D}_\pi(x_0) := \mathbb{E}\big[\sum_{t=0}^{\infty} \gamma^t d(x_t) \mid x_0, \pi\big]$ is the safety constraint function (expected cumulative constraint cost).

The goal in CMDPs is to solve the constrained optimization problem

$$\pi^* \in \min_{\pi \in \Delta} \left\{ \mathcal{C}_\pi(x_0) : \mathcal{D}_\pi(x_0) \leq d_0 \right\}. \tag{1}$$

Using the discounting property of a CMDP, Theorem 3.1 in Altman (1999) shows that if the feasibility set is non-empty, then there exists an optimal policy in the class of stationary Markovian policies $\Delta$. To motivate the CMDP formulation, we refer the reader to Chow et al. (2018), which presents two real-world examples in modeling safety using (i) the reachability constraint, and (ii) the constraint that limits the agent's visits to undesirable states.

## 3 A Lyapunov Approach for Solving CMDPs

In this section, we revisit the *Lyapunov approach* to solving CMDPs proposed by Chow et al. (2018) and report the mathematical results that are important in developing our safe policy optimization algorithms. To start, without loss of generality, we assume that we have access to a *baseline* feasible policy of Equation 1, $\pi_B$; i.e. $\pi_B$ satisfies $\mathcal{D}_{\pi_B}(x_0) \leq d_0$. We define a set of Lyapunov functions w.r.t. initial state $x_0 \in \mathcal{X}$ and constraint threshold $d_0$ as $\mathcal{L}_{\pi_B}(x_0, d_0) = \{L : \mathcal{X} \to \mathbb{R}_{\geq 0} : T_{\pi_B,d}[L](x) \leq L(x), \forall x \in \mathcal{X}; L(x_0) \leq d_0\}$, and call the constraints in this feasibility set *Lyapunov constraints*. For any arbitrary Lyapunov function $L \in \mathcal{L}_{\pi_B}(x_0, d_0)$, we denote by $\mathcal{F}_L(x) = \{\pi(\cdot|x) \in \Delta : T_{\pi,d}[L](x) \leq L(x)\}$ the set of $L$-induced Markov stationary policies. Since $T_{\pi,d}$ is a contraction mapping (Bertsekas, 2005), any $L$-induced policy $\pi$ has the property $\mathcal{D}_\pi(x) = \lim_{k \to \infty} T_{\pi,d}^k[L](x) \leq L(x), \forall x \in \mathcal{X}$. Together with the property that $L(x_0) \leq d_0$, they imply that any $L$-induced policy is a feasible policy of Equation 1. However, in general, the set $\mathcal{F}_L(x)$ does not necessarily contain an optimal policy of Equation 1, and thus it is necessary to design a Lyapunov function (w.r.t. a baseline policy $\pi_B$) that provides this guarantee. In other words, the main goal is to construct a Lyapunov function $L \in \mathcal{L}_{\pi_B}(x_0, d_0)$ such that

$$L(x) \geq T_{\pi^*,d}[L](x), \qquad L(x_0) \leq d_0. \tag{2}$$

Chow et al. (2018) show in their Theorem 1 that **1)** without loss of optimality, the Lyapunov function can be expressed as $L_\epsilon(x) := \mathbb{E}\left[\sum_{t=0}^\infty \gamma^t(d(x_t) + \epsilon(x_t)) \mid \pi_B, x\right]$, where $\epsilon(x) \geq 0$ is some auxiliary constraint cost uniformly upper-bounded by $\epsilon^*(x) := 2D_{\max}D_{TV}(\pi^*||\pi_B)(x)/(1 - \gamma)$, and **2)** if the baseline policy $\pi_B$ satisfies the condition $\max_{x \in \mathcal{X}} \epsilon^*(x) \leq D_{\max} \cdot \min\{(1 - \gamma)(d_0 - \mathcal{D}_{\pi_B}(x_0))/D_{\max}, D_{\max} - (1 - \gamma)\overline{\mathcal{D}}/D_{\max} + (1 - \gamma)\overline{\mathcal{D}}\}$, where $\overline{\mathcal{D}} = \max_{x \in \mathcal{X}} \max_\pi \mathcal{D}_\pi(x)$ is the maximum constraint cost, then the Lyapunov function candidate $L_{\epsilon^*}$ also satisfies the properties of Equation 2, and thus, its induced feasible policy set $\mathcal{F}_{L_{\epsilon^*}}$ contains an optimal policy. Furthermore, suppose that the distance between the baseline and optimal policies can be estimated effectively. Using the set of $L_{\epsilon^*}$-induced feasible policies and noting that the *safe* Bellman operator $T[V](x) = \min_{\pi \in \mathcal{F}_{L_{\epsilon^*}}(x)} T_{\pi,c}[V](x)$ is monotonic and contractive, one can show that $T[V](x) = V(x), \forall x \in \mathcal{X}$ has a unique fixed point $V^*$, such that $V^*(x_0)$ is a solution of Equation 1, and an optimal policy can be constructed via greedification, i.e., $\pi^*(\cdot|x) \in \arg\min_{\pi \in \mathcal{F}_{L_{\epsilon^*}}(x)} T_{\pi,c}[V^*](x)$. This shows that under the above assumption, Equation 1 can be solved using standard dynamic programming (DP) algorithms. While this result connects CMDP with Bellman's principle of optimality, verifying whether $\pi_B$ satisfies this assumption is challenging when a good estimate of $D_{TV}(\pi^*||\pi_B)$ is not available. To address this issue, Chow et al. (2018) propose to approximate $\epsilon^*$ with an auxiliary constraint cost $\widetilde{\epsilon}$, which is the *largest* auxiliary cost satisfying the Lyapunov condition $L_{\widetilde{\epsilon}}(x) \geq T_{\pi_B,d}[L_{\widetilde{\epsilon}}](x), \forall x \in \mathcal{X}$ and the safety condition $L_{\widetilde{\epsilon}}(x_0) \leq d_0$. The intuition here is that the larger $\widetilde{\epsilon}$, the larger the set of policies $\mathcal{F}_{L_{\widetilde{\epsilon}}}$. Thus, by choosing the largest such auxiliary cost, we hope to have a better chance of including the optimal policy $\pi^*$ in the set of feasible policies. Specifically, $\widetilde{\epsilon}$ is computed by solving the following linear programming (LP) problem:

$$\widetilde{\epsilon} \in \arg\max_{\epsilon : \mathcal{X} \to \mathbb{R}_{\geq 0}} \left\{ \sum_{x \in \mathcal{X}} \epsilon(x) : d_0 - \mathcal{D}_{\pi_B}(x_0) \geq \mathbf{1}(x_0)^\top (I - \gamma\{P(x'|x, \pi_B)\}_{x,x' \in \mathcal{X}})^{-1}\epsilon \right\}, \tag{3}$$

where $\mathbf{1}(x_0)$ represents a one-hot vector in which the non-zero element is located at $x = x_0$. When $\pi_B$ is a feasible policy, this problem has a non-empty solution. Furthermore, according to the derivations in Chow et al. (2018), the maximizer of Equation 3 is an indicator function of the form $\widetilde{\epsilon}(x) = (d_0 - \mathcal{D}_{\pi_B}(x_0)) \cdot \mathbf{1}\{x = \underline{x}\}/\mathbb{E}[\sum_{t=0}^\infty \gamma^t \mathbf{1}\{x_t = \underline{x}\} \mid x_0, \pi_B] \geq 0$, where $\underline{x} \in \arg\min_{x \in \mathcal{X}} \mathbb{E}\left[\sum_{t=0}^\infty \gamma^t \mathbf{1}\{x_t = x\} \mid x_0, \pi_B\right]$. They also show that by further restricting $\widetilde{\epsilon}(x)$ to be a constant function, the maximizer is given by $\widetilde{\epsilon}(x) = (1 - \gamma) \cdot (d_0 - \mathcal{D}_{\pi_B}(x_0)), \forall x \in \mathcal{X}$. Using the construction of the Lyapunov function $L_{\widetilde{\epsilon}}$, Chow et al. (2018) propose the safe policy iteration (SPI)

algorithm (see Algorithm 1 in Appendix A) in which the Lyapunov function is updated via *boot-strapping*, i.e., at each iteration $L_{\widetilde{\epsilon}}$ is recomputed using Equation 3 w.r.t. the current baseline policy. This algorithm has the following properties: **1)** *Consistent Feasibility*, i.e., if the current policy $\pi_k$ is feasible, then $\pi_{k+1}$ is also feasible; **2)** *Monotonic Policy Improvement*, i.e., $\mathcal{C}_{\pi_{k+1}}(x) \leq \mathcal{C}_{\pi_k}(x)$ for any $x \in \mathcal{X}$; and **3)** *Asymptotic Convergence*. Despite all these nice properties, SPI is still a value-function-based algorithm, and thus it is not straightforward to use it in continuous action problems. The main reason is that the greedification step becomes an optimization problem over the continuous set of actions that is not necessarily easy to solve. In Section 4, we show how we use SPI and its nice properties to develop safe policy optimization algorithms that can handle continuous action problems. Our algorithms can be thought as combinations of DDPG or PPO (or any other on-policy or off-policy policy optimization algorithm) with a SPI-inspired critic that evaluates the policy and computes its corresponding Lyapunov function. The computed Lyapunov function is then used to guarantee safe policy update, i.e., the new policy is selected from a restricted set of safe policies defined by the Lyapunov function of the current policy.

# 4    SAFE POLICY GRADIENT ALGORITHMS WITH LYAPUNOV FUNCTIONS

Policy gradient (PG) algorithms optimize a policy end-to-end by computing sample estimates of the gradient of the cumulative cost induced by the policy and then updating the policy in the gradient direction. In general, stochastic policies that give a probability distribution over actions are parameterized by a $\kappa$-dimensional vector $\theta$, so the space of policies can be written as $\left\{\pi_\theta(\cdot|x), x \in \mathcal{X}, \theta \in \mathbb{R}^\kappa\right\}$. Since in this setting a policy $\pi$ is uniquely defined by its parameter vector $\theta$, policy-dependent functions can be written as a function of $\theta$ or $\pi$, and they are used interchangeably in this paper.

Recently there are two PG algorithms which have emerged as generally well-performing, namely DDPG and PPO. These algorithms are widely used in many continuous control tasks. DDPG (Lillicrap et al., 2015) is an off-policy Q-learning style algorithm. A deterministic policy $\pi_\theta(x)$ and a Q-value approximator $Q(x, a; \phi)$ are trained jointly. The Q-value approximator is trained to minimize Bellman errors with respect to $\pi$; i.e., it is trained to fit the true Q-value function satisfying $Q(x, a) = c(x, a) + \gamma \sum_{x' \in \mathcal{X}} P(x'|x, a)Q(x', \pi_\theta(x'))$. The policy $\pi$ is then trained to optimize $Q(x, \pi_\theta(x); \phi)$ via chain-rule. The PPO algorithm we use is a penalty form of TRPO (Schulman et al., 2017) with an adaptive rule to tune the $D_{KL}$ penalty weight $\beta_k$. Specifically, PPO trains a Gaussian policy $\pi_\theta(x)$ according to the standard policy gradient objective augmented with a penalty on KL-divergence from a previous version of the policy; i.e., the penalty is of the form, $\overline{D}_{\mathrm{KL}}(\theta, \theta') = \mathbb{E}[D_{\mathrm{KL}}(\pi_{\theta'}(\cdot|x_t)||\pi_\theta(\cdot|x_t))|x_0, \theta']$, where $\theta'$ is a previous version of the policy.

In CMDPs, the presence of a constraint $\mathcal{D}_{\pi_\theta}(x_0) \leq d_0$ may be naively incorporated into the standard forms of DDPG and PPO via the Lagrangian method. That is, one may transform the constrained optimization problem to a penalty form, in which the constraint costs $d(x)$ are added to the task costs $c(x, a)$. The resulting penalized form of the objective is $\min_\theta \max_{\lambda \geq 0} \mathbb{E}\left[\sum_{t=0}^\infty c(x_t, a_t) + \lambda d(x_t)|x_0, \theta\right] - \lambda d_0$. In this form, both $\theta$ and $\lambda$ must be optimized jointly to find a saddle-point of the objective. The optimization of $\theta$ may be performed by either DDPG or PPO on the augmented cost $c(x, a) + \lambda d(x)$. The optimization of $\lambda$ may be performed by stochastic gradient descent on $x_t$ taken from trajectories sampled according to $\theta$.

Although the Lagrangian approach is easy to implement (see Appendix B for details), in practice it does not lead to safety in training. While the objective encourages finding a solution which is safe, any intermediate step in the optimization may lead to an unsafe policy. In contrast, the Lyapunov approaches we propose are guaranteed to return a safe policy, not only at convergence, but also during training. In the subsections below, we elaborate how to transform DDPG and PPO to their Lyapunov safe counterparts below:

$$\theta^* = \arg\min_\theta \mathcal{C}_{\pi_\theta}(x_0) \quad \text{subject to} \quad \int_{a \in \mathcal{A}} (\pi_\theta(a|x) - \pi_B(a|x))Q_L(x, a)da \leq \widetilde{\epsilon}(x), \forall x \in \mathcal{X}. \quad (4)$$

where $Q_L(x, a) = d(x) + \widetilde{\epsilon}'(x) + \gamma \sum_{x'} P(x'|x, a)L_{\widetilde{\epsilon}'}(x')$ is the state-action Lyapunov function.

We will describe two approaches to incorporating Lyapunov constraints in PG: $\theta$-projection and $a$-projection. In Section 4.1, we formulate the Lyapunov-based PG using constrained policy optimization (which we call $\theta$-projection), and in Section 4.2 we show how the Lyapunov constraints can be embedded into the policy network via a safety layer (which we call $a$-projection). In Section A.1, we also discuss two practical techniques to further enforce safety during policy training.

## 4.1 Constrained Optimization Approach to Lyapunov-based PG

For demonstration purposes, we hereby show how one can perform constrained policy optimization with PPO and Lyapunov constraints. With almost identical machinery, this procedure can also be applied to DDPG. Consider the following constrained optimization at iteration $k$ with *semi-infinite dimensional* Lyapunov constraints for policy update:

$$\theta \in \underset{\theta \in \Theta}{\arg\min} \quad \langle(\theta - \theta_k), \nabla_\theta \mathbb{E}_{x \sim \mu_{\theta_k}, a \sim \pi_\theta} \left[Q_{\theta_k}(x, a)\right] + \beta_k \langle(\theta - \theta_k), \nabla_\theta^2 D_{\text{KL}}(\theta \| \theta_k) \mid_{\theta = \theta_k} \cdot (\theta - \theta_k) \mid_{\theta = \theta_k}\rangle$$

$$\text{s.t.} \quad \langle(\theta - \theta_k), \nabla_\theta \mathbb{E}_{a \sim \pi_\theta} \left[Q_{L_{\theta_k}}(x, a)\right] \mid_{\theta = \theta_k}\rangle \leq \widetilde{\epsilon}(x), \ \forall x \in \mathcal{X},$$

where $\mu_{\theta_k}$ is the $\gamma$-visiting distribution w.r.t. $\pi_{\theta_k}$, and $\beta_k$ is the adaptive penalty weight of the $\overline{D}_{\text{KL}}(\theta \| \theta_k)$ regularizer. Clearly if one updates the policy parameter by solving the above optimization, and if the approximation errors from neural network parameterizations of $Q_{\theta_k}$ and $Q_{L_{\theta_k}}$ and first-order Taylor series expansion are small, then safety may ensure safety during training. However, the presence of infinite-dimensional Lyapunov constraints makes solving the above optimization (in real-time) numerically intractable. To tackle the issue of infinite dimensionality, (without loss of optimality) we re-write the Lyapunov constraint in the following form: $\max_{x \in \mathcal{X}} \langle(\theta - \theta_k), \nabla_\theta \mathbb{E}_{a \sim \pi_\theta} \left[Q_{L_{\theta_k}}(x, a)\right] \mid_{\theta = \theta_k} - \widetilde{\epsilon}(x) \leq 0$. This might still lead to numerical instability in gradient descent algorithms, because the $\max$-operator in the constraint is non-differentiable. Similar to the surrogate constraint used in TRPO (to transform the $\max D_{\text{KL}}$ constraint into an average $\overline{D}_{\text{KL}}$ constraint), a more numerically stable way is to *approximate* the Lyapunov constraint using the following average constraint surrogate:

$$\langle(\theta - \theta_k), \frac{1}{M} \sum_{i=1}^{M} \nabla_\theta \mathbb{E}_{a \sim \pi_\theta} \left[Q_{L_{\theta_k}}(x_i, a)\right] \mid_{\theta = \theta_k}\rangle \leq \frac{1}{M} \sum_{i=1}^{M} \widetilde{\epsilon}(x_i). \tag{5}$$

where $N$ is the number of on-policy trajectories of $\pi_{\theta_k}$. In practice, if one adopts $\widetilde{\epsilon} = (1 - \gamma)(d_0 - \mathcal{D}_{\pi_{\theta_k}}(x_0))$ from Section 4.1, then the linear term in Equation 5 can be simplified as $\nabla_\theta \mathbb{E}_{a \sim \pi_\theta} \left[Q_{L_{\theta_k}}(x_i, a)\right] = \nabla_\theta \int_{a \in \mathcal{A}} \pi_\theta(a|x) \nabla_\theta \log \pi_\theta(a|x) Q_{D,\theta_k}(x_i, a) da$. On the other hand, by setting $M = O(1/(1 - \gamma))$, the constraint threshold becomes $\frac{1}{M} \sum_{i=1}^{M} \widetilde{\epsilon}(x_i) \approx d_0 - \mathcal{D}_{\pi_{\theta_k}}(x_0)$. Collectively, the average constraint surrogate in Equation 5 becomes $\langle(\theta - \theta_k), \frac{1}{M} \sum_{i=1}^{M} \nabla_\theta \mathbb{E}_{a \sim \pi_\theta} \left[Q_{D,\theta_k}(x_i, a)\right] \mid_{\theta = \theta_k}\rangle \leq d_0 - \mathcal{D}_{\pi_{\theta_k}}(x_0)$, which is equivalent to the constraint used in the CPO algorithm (see Section 6.1 in Achiam et al. (2017)). This draws the connection between CPO and Lyapunov-based PG with $\theta$-projection.

The Lyapunov-based algorithms with $\theta$-projection in constrained policy update is summarized by Algorithm 4 in Appendix A. In the experiment section, we denote the DDPG version and the PPO version of this algorithm by SDDPG and SPPO respectively.

## 4.2 Embedding Lyapunov Constraints into a Safety Layer

Notice that the main contribution of the Lyapunov approach is to break down a trajectory-based constraint into a sequence of single-step, *state dependent* constraints. When the state space is infinite/continuous, it is counter-intuitive to directly enforce these Lyapunov constraints (instead of the original trajectory-based constraint) in the optimization w.r.t. policy parameter $\theta$, because the feasibility set is characterized by infinite dimensional constraints. Rather, leveraging the ideas of a *safety layer* from Dalal et al. (2018) that was applied to single-step constraints, we propose a novel approach to embed the set of Lyapunov constraints into the policy network. In this way, one reformulates the CMDP problem into an unconstrained one, whose policy parameter $\theta$ (of the augmented network) can then be optimized by any standard unconstrained PG algorithms. At every given state, the unconstrained action is first computed and is then passed through the safety layer, where a feasible action mapping is constructed by projecting the unconstrained actions onto the feasibility set w.r.t. the corresponding Lyapunov constraint. Therefore, safety during training w.r.t. the original CMDP problem is guaranteed by the Lyapunov theorem.

We hereby demonstrate how one can find a feasible action mapping using the safety layer with DDPG, whose role is to solve the following projection problem at given state $x \in \mathcal{X}$:

$$a^*(x) \in \underset{a}{\arg\min} \left\{\frac{1}{2}\|a - \pi_{\theta,\text{unc}}(x)\|^2 : (a - \pi_B(x))^\top \nabla_a Q_L(x, a) \mid_{a = \pi_B(x)} \leq \widetilde{\epsilon}(x)\right\}. \tag{6}$$

In the above optimization problem, $\pi_{\theta,\text{unc}}$ is the unconstrained policy whose policy parameter is updated by standard DDPG, $\pi_B$ is the current data-generation policy that is safe, and the left side of the

constraint is the first-order Taylor series approximation of $\int_{a \in \mathcal{A}} Q_L(x, \pi_\theta(x)) - Q_L(x, \pi_B(x))da$ over actions, w.r.t. the action $\pi_B(x)$ induced by the baseline policy. As in Section 4.1, since the auxiliary cost $\widetilde{\epsilon}$ is state-dependent, one can readily find $\nabla_a Q_L(x, a) \mid_{a=\pi_B(x)}$ by computing the gradient of the constraint action value function $\nabla_a Q_D(x, a) \mid_{a=\pi_B(x)}$. Generally, the safety layer perturbs the unconstrained action as little as possible in the Euclidean norm in order to satisfy the Lyapunov constraints. Notice that the objective function is positive-definite and quadratic and the constraint approximation is linear. Therefore, we can find the global solution to this convex problem effectively via an in-graph iterative QP-solver, such as the one from Amos & Kolter (2017). Furthermore, since the optimization in Equation 6 only has a single Lyapunov constraint, one can express $a^*(x)$ using the following analytical solution.

**Proposition 1.** *At any given state $x \in \mathcal{X}$, the solution to the optimization problem in Equation 6 has the following form:* $a^*(x) = \pi_{\theta, unc}(x) + \lambda^*(x) \nabla_a Q_L(x, a) \mid_{a=\pi_B(x)}$, *where*

$$\lambda^*(x) = \left( \left( \left( \nabla_a Q_L(x, a) \mid_{a=\pi_B(x)} \right)^\top (\pi_{\theta, unc}(x) - \pi_B(x)) - \widetilde{\epsilon}(x) \right) \Big/ \left( \nabla_a Q_L(x, a) \mid_{a=\pi_B(x)} \right)^\top \nabla_a Q_L(x, a) \mid_{a=\pi_B(x)} \right)_+ .$$

The closed-form solution is essentially a linear projection of the unconstrained action $\pi_{\theta, \text{unc}}(x)$ to the safe hyperplane characterized with slope $\nabla_a Q_L(x, a) \mid_{a=\pi_B(x)}$ and intercept $\widetilde{\epsilon}(x) = (1 - \gamma)(d_0 - \mathcal{D}_{\pi_B}(x_0))$. Implementing $a^*(x)$ is very simple; it only consists of several arithmetic operations such as matrix products and ReLU. Extending this closed-form solution to handle multiple constraints is possible, under the assumption of having at most one constraint active at a time.

In general, the safety layer approach can also be applied to policy gradient algorithms, such as PPO, that learn a non-deterministic policy. For example in the PPO case when the policy is parameterized with a Gaussian distribution, then one simply need to project both the mean and the standard-deviation vector, in order to obtain a feasible action probability. The Lyapunov-based algorithms with $a$-projection in safety layer is summarized by Algorithm 5 in Appendix A. In the experiment section, we denote the DDPG version and the PPO version of this algorithm by SDDPG-modular and SPPO-modular respectively.

## 5 Experiments

We empirically validate the Lyapunov-based PG algorithms on several robot locomotion continuous control tasks. In these experiments we aim to address the following questions about our proposed algorithm: (i) How is the performance (in terms of cost and safety during training) of Lyapunov-based PG compared to other baseline methods such as CPO and the naive Lagrangian approach? (ii) In the presence of approximation errors in value functions and policies, how robust is Lyapunov-based PG algorithm with respect to constraint violations?

To understand the performance of these algorithms in terms of both cost and safety guarantees, we designed several interpretable experiments in simulated robot locomotion continuous control tasks, whose notions of safety are motivated by physical constraints. We consider three domains using the MuJoCo simulator (Todorov et al., 2012) with a variety of different agents: (i)HalfCheetah-Safe: The HalfCheetah agent is rewarded for running, but its speed is limited for stability and safety; (ii) Point-Circle: The Point agent is rewarded for running in a wide circle, but is constrained to stay within a safe region defined by $|x| \leq x_{\text{lim}}$ (Achiam et al., 2017); (iii) Point-Gather & Ant-Gather: The agent, which is either a Point or an Ant, is rewarded for collecting target objects in a terrain map, while being constrained to avoid bombs (Achiam et al., 2017).

Visualizations of these tasks as well as more detailed descriptions (immediate cost and constraint cost functions, constraint thresholds) are given in Appendix C. In these experiments there are three different agents: (1) a point-mass ($\mathcal{X} \subseteq \mathbb{R}^9, A \subseteq \mathbb{R}^2$); an ant quadruped robot ($\mathcal{X} \subseteq \mathbb{R}^{32}, A \subseteq \mathbb{R}^8$); (3) a half-cheetah ($\mathcal{X} \subseteq \mathbb{R}^{18}, A \subseteq \mathbb{R}^6$). For all experiments, we use two neural networks with two hidden layers of size $(100, 50)$ and ReLU activations to model the mean and log-variance of the Gaussian actor policy, and two neural networks with two hidden layers of size $(200, 50)$ and $\tanh$ activations to model the critic and constraint critic. To build a low variance sample gradient estimate, we use GAE-$\lambda$ (Schulman et al., 2015b) to estimate the advantage and constraint advantage functions, with a hyper-parameter $\lambda \in (0, 1)$ optimized by grid-search.

### 5.1 Comparison with Unconstrained PG, Lagrangian Approach, and CPO

For comparison purposes, we implement the naive Lagrangian approach. Details of this algorithm are available in Appendix B. For fair comparisons, we also optimize the Lagrangian using a natural

policy gradient when needed. To understand the optimal unconstrained performance on these environments, we also include the learning performance of two state-of-the-art unconstrained reinforcement learning algorithms, namely DDPG (Lillicrap et al., 2015) and PPO (Schulman et al., 2017). In the PPO case, we aim to compare the performance of CPO (which is equivalent to Lyapunov-based PG with $\theta$-projection) with Lyapunov-based PG (with $a$-projection) . Notice that the original implementation of CPO in Achiam et al. (2017) is based on TRPO (Schulman et al., 2015a). Instead of the original CPO algorithm, we use its PPO alternative (which coincides with the SPPO algorithm derived in Section 4.1) as the safe RL baseline for comparison. SPPO preserves the essence of CPO by adding a 1st order constraint to the proximal policy optimization. The main difference between CPO and SPPO is that the latter does not perform backtracking line-search. The decision to compare with SPPO instead of CPO is 1) to avoid the additional computational complexity of line-search in TRPO, while maintaining the performance of PG using the popular PPO algorithm, 2) to have a back-propagatable version of CPO, and 3) to have a fair comparison with other back-propagatable safe RL algorithms, such as the DDPG and safety layer counterparts.

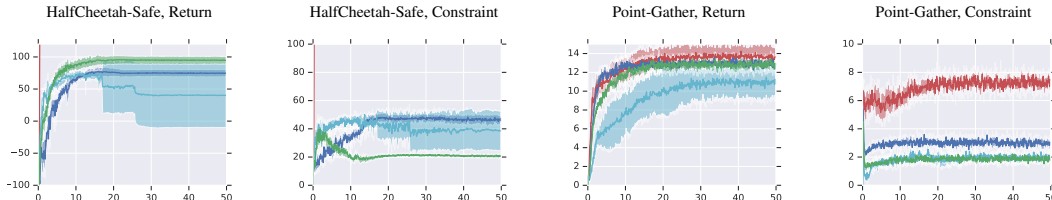

Figure 1: Results of various DDPG algorithms on safe robot locomotion tasks, with x-axis in thousands of episodes. We include runs from DDPG (red), DDPG-Lagrangian (magenta), SD-DPG (blue), SDDPG-modular (green) on HalfCheetah-Safe and Point-Gather. We discover that the Lyapunov-based approaches can perform safe learning, despite the fact that the environment dynamics model and cost functions are not known, control actions are continuous, and deep function approximations are necessary.

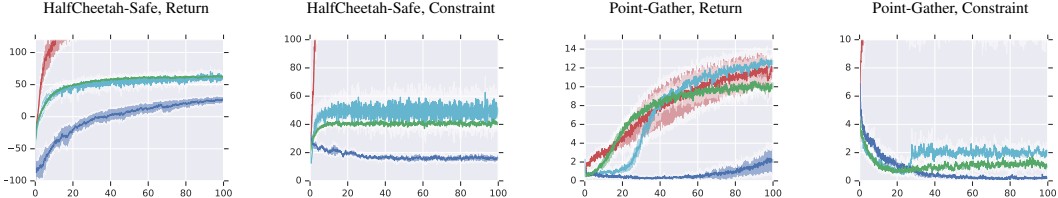

Figure 2: Results of various PPO algorithms on safe robot locomotion tasks, with x-axis in thousands of episodes. We include runs from PPO (red), PPO-Lagrangian (magenta), SPPO (blue), SPPO-modular (green) on HalfCheetah-Safe and Point-Gather. Similar to Figure 1, the Lyapunov-based approaches can perform safe learning in the control tasks when function approximations on policies and value functions are necessary.

Learning curves for unconstrained DDPG, Lagrangian DDPG, SDDPG and SDDPG-modular are shown in Figure 1, and the learning curves for unconstrained PPO, Lagrangian PPO, SPPO and SPPO-modular are shown in Figure 6. Due to space restrictions, more results are included in Appendix C. From the comparison plots, one can clearly see that the unconstrained DDPG and PPO agents are constraint-violating in these environments. Although the Lagrangian approaches in DDPG and PPO converge to feasible policies with reasonably good performance, in accord with our earlier claims these algorithms cannot guarantee constraint satisfaction during training. Furthermore it is worth-noting that the Lagrangian approach can be sensitive to the initialization of the Lagrange multiplier $\lambda_0$. If $\lambda_0$ is too large, it would make policy updates overly conservative, while if $\lambda_0$ is too small then constraint violation will be more pronounced. By default, we initialize $\lambda_0 = 10$, which assumes no knowledge about the environment.

Generally in both DDPG and PPO experiments, the Lyapunov-based PG algorithms lead to more stable learning and constraint satisfaction than the Lagrangian approach. The Lyapunov approaches quickly stabilize the constraint cost to be below the threshold, while the constraint costs from the Lagrangian approach tend to jiggle around the threshold. In many cases the $a$-projection Lyapunov-based PG (SDDPG-modular, SPPO-modular) converges faster than the $\theta$-projection counterpart (SDDPG, SPPO). This corroborates with the hypothesis that the $a$-projection approach is less conservative during policy updates than the $\theta$-projection approach (which is what CPO is based on).

Finally, in most experiments (HalfCheetah, PointGather, and AntGather) the DDPG class of algorithms tends to have faster learning than the PPO counterpart. This is potentially due to the improved data-efficiency when using off-policy samples in PG updates. Although this benefit is not directly related to the addition of Lyapunov constraints, this also supports our claim that some Lyapunov-based safe PG algorithms (SDDPG, SDDPG-modular) can learn more effectively than CPO (which is analogous to SPPO).

## 5.2 Constraint Violation During Training

Due to function approximation errors in policies and value functions, in practice most safe learning algorithms including the Lyapunov-based PG methods may still take a bad step and lead to constraint violation. While methods like safeguard policy update and constraint tightening from Section A.1 might help to remedy this issue, it is still unclear how robust each algorithm is regarding constraint satisfaction during training. To study the degree of constraint violation in different PG algorithms, we show their constraint violation plots in Figure 3 for DDPG-based and PPO-based algorithms. To make the comparison fair, we apply both safeguard policy update and the constraint tightening to all safe PG algorithms. From the constraint violation plots, it is clear that the Lagrangian methods (both DDPG and PPO) violate safety constraints more often than the Lyapunov-based counterparts. In many cases the Lyapunov-based PG algorithms with $a$-projection have lower constraint violation than its $\theta$-projection counterparts. We speculate this is due to the fact that the safety layer generates smoother gradient updates during end-to-end training.

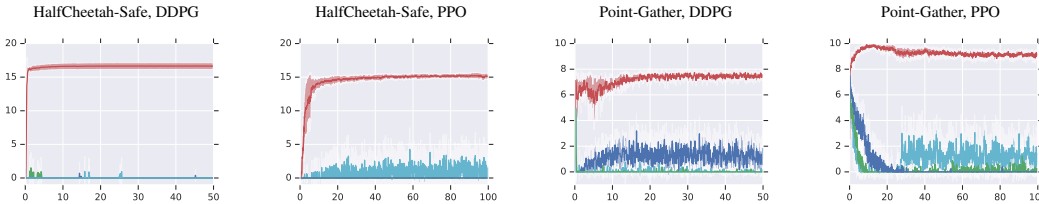

Figure 3: Constraint violations (logarithmic scale, i.e., $\log(1 + \text{ccv})$, where ccv represents the area under constraint learning curve that is above the constraint threshold) of various PG algorithms on safe robot locomotion tasks, with x-axis in thousands of episodes. We include runs from DDPG (red), DDPG-Lagrangian (magenta), SDDPG (blue), SDDPG-modular (green), PPO (red), PPO-Lagrangian (magenta), SPPO (blue), SPPO-modular (green) on HalfCheetah-Safe and Point-Gather. Compared with Lagrangian approach, we discover that the Lyapunov-based approaches generally have more stable and safe learning, which lead to lower constraint violations.

## 6 Conclusions

In this paper, we formulated the problem of safe RL as a CMDP and used the notion of Lyapunov function to develop policy optimization algorithms that learn policies that are both safe and have low expected cumulative cost. Our algorithms extend the Lyapunov-based approach to solving CMDPs of Chow et al. (2018) to continuous action problems. Our algorithms combine DDPG or PPO (or any other on-policy or off-policy policy optimization algorithm) with a critic that is inspired by the safe policy iteration (SPI) algorithm of Chow et al. (2018) and both evaluates the policy and computes its corresponding Lyapunov function. The computed Lyapunov function is then used to guarantee safe policy update. We can categorize our algorithms into two classes in terms of the way they perform safe policy update. The first class is based on constrained optimization w.r.t. policy parameter, similar to what is done in CPO. The second class relies on the safety layer concept (Dalal et al., 2018) that integrates the Lyapunov constraints into the policy network by adding an action projection layer to it. We evaluated our algorithms on four high-dimensional simulated robot locomotion tasks and compared them with CPO (Achiam et al., 2017) and the Lagrangian method in terms of minimizing the expected cumulative return and constraint violation during training. Our results indicate that our Lyapunov-based algorithms **1)** achieve safe learning, **2)** have better data-efficiency, and **3)** can be more naturally integrated within the standard end-to-end differentiable policy gradient training pipeline. In general, our work is a step forward in deploying RL to real-world problems in which safety guarantees are of paramount importance.

Future work includes **1)** developing more stable and safe learning algorithms by further exploiting the properties of Lyapunov functions, **2)** exploring more efficient ways to include Lyapunov constraints in constrained policy optimization, and **3)** applying the Lyapunov-based PG algorithms to real-world continuous control problems, particularly in robotics.

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

## A DETAILS OF THE SAFE POLICY GRADIENT ALGORITHMS

---

**Algorithm 1** Safe Policy Iteration (SPI)

---

**Input:** Initial feasible policy $\pi_0$;
**for** $k = 0, 1, 2, \ldots$ **do**
    **Step 0:** With $\pi_b = \pi_k$, evaluate the Lyapunov function $L_{\epsilon_k}$, where $\epsilon_k$ is a solution of Equation 3
    **Step 1:** Evaluate the cost value function $V_{\pi_k}(x) = \mathcal{C}_{\pi_k}(x)$; Then update the policy by solving the
    following problem: $\pi_{k+1}(\cdot|x) \in \operatorname{argmin}_{\pi \in \mathcal{F}_{L_{\epsilon_k}}(x)} T_{\pi,c}[V_{\pi_k}](x), \forall x \in \mathcal{X}$
**end for**
**Return** Final policy $\pi_{k^*}$

---

**Algorithm 2** Trajectory-based Policy Gradient Algorithm for CMDP

---

**Input:** parameterized policy $\pi(\cdot|\cdot; \theta)$
**Initialization:** policy parameter $\theta = \theta_0$, and the Lagrangian parameter $\lambda = \lambda_0$
**while** TRUE **do**
    **for** $i = 0, 1, 2, \ldots$ **do**
        **for** $j = 1, 2, \ldots$ **do**
            Generate $N$ trajectories $\{\xi_{j,i}\}_{j=1}^N$ by starting at $x_0$ and following the policy $\theta_i$.
        **end for**

$$\theta \text{ Update:} \quad \theta_{i+1} = \theta_i - \alpha_{2,i} \frac{1}{N} \sum_{j=1}^N \nabla_\theta \log \mathbb{P}_\theta(\xi_{j,i})|_{\theta=\theta_i} \left( \mathcal{C}(\xi_{j,i}) + \lambda_i \mathcal{D}(\xi_{j,i}) \right)$$

$$\lambda \text{ Update:} \quad \lambda_{i+1} = \Gamma_\Lambda \left[ \lambda_i + \alpha_{1,i} \left( -d_0 + \frac{1}{N} \sum_{j=1}^N \mathcal{D}(\xi_{j,i}) \right) \right]$$

    **end for**
    **if** $\{\lambda_i\}$ converges to $\lambda_{\max}$ **then**
        Set $\lambda_{\max} \leftarrow 2\lambda_{\max}$.
    **else**
        **return** parameters $\nu, \theta, \lambda$ and **break**
    **end if**
**end while**

---

**Algorithm 3** Actor-Critic Algorithms for CMDP

---

**Input:** Parameterized policy $\pi(\cdot|\cdot; \theta)$ and value function feature vector $\phi(\cdot)$
**Initialization:** policy parameters $\theta = \theta_0$; Lagrangian parameter $\lambda = \lambda_0$; value function weight $v = v_0$
**while** TRUE **do**
    **for** $k = 0, 1, 2, \ldots$ **do**
        Sample $a_k \sim \pi(\cdot|x_k; \theta_k)$; $C_{\lambda_k}(x_k, a_k) = C(x_k, a_k) + \lambda_k D(x_k, a_k)$; $x_{k+1} \sim P(\cdot|x_k, a_k)$;
        // AC Algorithm:

$$\text{TD Error:} \quad \delta_k(v_k) = C_{\lambda_k}(x_k, a_k) + \gamma \widehat{V}_{\phi_k}(x_{k+1}) - \widehat{V}_{\phi_k}(x_k) \tag{7}$$

$$\text{Critic Update:} \quad v_{k+1} = v_k + \zeta_3(k)\delta_k(v_k)\psi(x_k) \tag{8}$$

$$\theta \text{ Update:} \quad \theta_{k+1} = \theta_k - \zeta_2(k)\nabla_\theta \log \pi_\theta(a_k|x_k) \cdot \delta_k(v_k)/1 - \gamma \tag{9}$$

$$\lambda \text{ Update:} \quad \lambda_{k+1} = \Gamma_\Lambda \left( \lambda_k + \zeta_1(k) \left( -d_0 + \frac{1}{N} \sum_{j=1}^N \mathcal{D}(\xi_{j,i}) \right) \right) \tag{10}$$

        // NAC Algorithm:

$$\text{Critic Update:} \quad w_{k+1} = \left( I - \zeta_3(k)\nabla_\theta \log \pi_\theta(a_k|x_k)|_{\theta=\theta_k} \left( \nabla_\theta \log \pi_\theta(a_k|x_k)|_{\theta=\theta_k} \right)^\top \right) w_k$$

$$+ \zeta_3(k)\delta_k(v_k)\nabla_\theta \log \pi_\theta(a_k|x_k)|_{\theta=\theta_k} \tag{11}$$

$$\theta \text{ Update:} \quad \theta_{k+1} = \theta_k - \zeta_2(k)w_k/1 - \gamma \tag{12}$$

        **Other Updates:** Follow from Eqs. 7, 8, and 10.

    **end for**
**end while**

---

### A.1 PRACTICAL IMPLEMENTATIONS OF SAFE PG

Due to function approximation errors, even with the Lyapunov constraints in practice the safe PG algorithm may take a bad step and produce an infeasible policy update and cannot automatically recover from such a bad step. To tackle this issue, similar to Achiam et al. (2017)

---

**Algorithm 4** Lyapunov-based Policy Gradient with $\theta$-projection (SDDPG and SPPO)

---

**Input:** Initial feasible policy $\pi_0$;

**for** $k = 0, 1, 2, \ldots$ **do**

**Step 0:** With $\pi_b = \pi_{\theta_k}$, generate $N$ trajectories $\{\xi_{j,k}\}_{j=1}^N$ of $T$ steps by starting at $x_0$ and following the policy $\theta_k$

**Step 1:** Using the trajectories $\{\xi_{j,k}\}_{j=1}^N$, estimate the critic $Q_\theta(x, a)$ and the constraint critic $Q_{D,\theta}(x, a)$;

- For DDPG, these functions are trained by minimizing the MSE of Bellman residual, and one can also use off-policy samples from replay buffer (Schaul et al., 2015);
- For PPO these functions can be estimated by the generalized advantage function technique from Schulman et al. (2015b)

**Step 2:** Based on the closed form solution of a QP problem with an LP constraint in Section 10.2 of Achiam et al. (2017), calculate $\lambda_k^*$ with the following formula:

$$\lambda_k^* = \left( \frac{-\beta_k \widetilde{\epsilon} - \left( \nabla_\theta Q_\theta(\bar{x}, \bar{a}) \mid_{\theta=\theta_k} \right)^\top H(\theta_k)^{-1} \nabla_\theta Q_{D,\theta}(\bar{x}, \bar{a}) \mid_{\theta=\theta_k}}{\left( \nabla_\theta Q_{D,\theta}(\bar{x}, \bar{a}) \mid_{\theta=\theta_k} \right)^\top H(\theta_k)^{-1} \nabla_\theta Q_{D,\theta}(\bar{x}, \bar{a}) \mid_{\theta=\theta_k}} \right)_+ ,$$

where

$$\nabla_\theta Q_\theta(\bar{x}, \bar{a}) = \frac{1}{N} \sum_{x,a \in \xi_{j,k}, 1 \leq j \leq N} \sum_{t=0}^{T-1} \gamma^t \nabla_\theta \log \pi_\theta(a|x) Q_\theta(x, a),$$

$$\nabla_\theta Q_{D,\theta}(\bar{x}, \bar{a}) = \frac{1}{N} \sum_{x,a \in \xi_{j,k}, 1 \leq j \leq N} \sum_{t=0}^{T-1} \gamma^t \nabla_\theta \log \pi_\theta(a|x) Q_\theta(x, a),$$

$\beta_k$ is the adaptive penalty weight of the $\overline{D}_{\text{KL}}(\pi||\pi_{\theta_k})$ regularizer, and $H(\theta_k) = \nabla_\theta^2 \overline{D}_{\text{KL}}(\pi||\pi_\theta) \mid_{\theta=\theta_k}$ is the Hessian of this term

**Step 3:** Update the policy parameter by following the objective gradient;

- For DDPG

$$\theta_{k+1} \leftarrow \theta_k - \alpha_k \cdot \frac{1}{N \cdot T} \sum_{x \in \xi_{j,k}, 1 \leq j \leq N} \nabla_\theta \pi_\theta(x) \mid_{\theta=\theta_k} \cdot (\nabla_a Q_{\theta_k}(x, a) + \lambda_k^* \nabla_a Q_{D,\theta_k}(x, a)) \mid_{a=\pi_{\theta_k}(x)}$$

- For PPO,

$$\theta_{k+1} \leftarrow \theta_k - \frac{\alpha_k}{N \beta_k} \left( H(\theta_k) \right)^{-1} \sum_{x_{j,t}, a_{j,t} \in \xi_{j,k}, 1 \leq j \leq N} \sum_{t=0}^{T-1} \gamma^t \cdot \nabla_\theta \log \pi_\theta(a_{j,t}|x_{j,t}) \mid_{\theta=\theta_k} \cdot$$
$$(Q_{\theta_k}(x_{j,t}, a_{j,t}) + \lambda_k^* Q_{D,\theta_k}(x_{j,t}, a_{j,t}))$$

**Step 4:** At any given state $x \in \mathcal{X}$, compute the feasible action probability $a^*(x)$ via action projection in the safety layer, that takes inputs $\nabla_a Q_L(x, a) = \nabla_a Q_{D,\theta_k}(x, a)$ and $\epsilon(x) = (1 - \gamma)(d_0 - Q_{D,\theta_k}(x_0, \pi_k(x_0)))$, for any $a \in \mathcal{A}$.

**end for**

**Return** Final policy $\pi_{\theta_{k^*}}$,

---

---

**Algorithm 5** Lyapunov-based Policy Gradient with $a$-projection (SDDPG-modular and SPPO-modular)

---

**Input:** Initial feasible policy $\pi_0$;

**for** $k = 0, 1, 2, \ldots$ **do**

    **Step 0:** With $\pi_b = \pi_{\theta_k}$, generate $N$ trajectories $\{\xi_{j,k}\}_{j=1}^N$ of $T$ steps by starting at $x_0$ and following the policy $\theta_k$

    **Step 1:** Using the trajectories $\{\xi_{j,k}\}_{j=1}^N$, estimate the critic $Q_\theta(x, a)$ and the constraint critic $Q_{D,\theta}(x, a)$;

       • For DDPG, these functions are trained by minimizing the MSE of Bellman residual, and one can also use off-policy samples from replay buffer (Schaul et al., 2015);

       • For PPO these functions can be estimated by the generalized advantage function technique from Schulman et al. (2015b)

    **Step 2:** Update the policy parameter by following the objective gradient;

       • For DDPG

$$\theta_{k+1} \leftarrow \theta_k - \alpha_k \cdot \frac{1}{N \cdot T} \sum_{x \in \xi_{j,k}, 1 \leq j \leq N} \nabla_\theta \pi_\theta(x) \mid_{\theta=\theta_k} \cdot \nabla_a Q_{\theta_k}(x, a) \mid_{a=\pi_{\theta_k}(x)};$$

       • For PPO,

$$\theta_{k+1} \leftarrow \theta_k - \frac{\alpha_k}{N\beta_k} \left( H(\theta_k) \right)^{-1} \sum_{x_{j,t}, a_{j,t} \in \xi_{j,k}, 1 \leq j \leq N} \sum_{t=0}^{T-1} \gamma^t \cdot \nabla_\theta \log \pi_\theta(a_{j,t}|x_{j,t}) \mid_{\theta=\theta_k} \cdot Q_{\theta_k}(x_{j,t}, a_{j,t})$$

       where $\beta_k$ is the adaptive penalty weight of the $\overline{D}_{\mathrm{KL}}(\pi || \pi_{\theta_k})$ regularizer, and $H(\theta_k) = \nabla_\theta^2 \overline{D}_{\mathrm{KL}}(\pi || \pi_\theta) \mid_{\theta=\theta_k}$ is the Hessian of this term

    **Step 3:** At any given state $x \in \mathcal{X}$, compute the feasible action probability $a^*(x)$ via action projection in the safety layer, that takes inputs $\nabla_a Q_L(x, a) = \nabla_a Q_{D,\theta_k}(x, a)$ and $\epsilon(x) = (1 - \gamma)(d_0 - Q_{D,\theta_k}(x_0, \pi_k(x_0)))$, for any $a \in \mathcal{A}$.

**end for**

**Return** Final policy $\pi_{\theta_{k^*}}$,

---

we propose the following *safeguard* policy update rule to purely decrease the constraint cost: $\theta_{k+1} = \theta_k - \alpha_{\mathrm{sg},k} \nabla_\theta \mathcal{D}_{\pi_\theta}(x_0)_{\theta=\theta_k}$, where $\alpha_{\mathrm{sg},k}$ is the learning rate for safeguard update. If $\alpha_{\mathrm{sg},k} >> \alpha_k$ (learning rate of PG), then with the safeguard update $\theta$ will quickly recover from the bad step but it might be overly conservative. This approach is principled because as soon as $\pi_{\theta_k}$ is unsafe/infeasible w.r.t. CMDP, the algorithm uses a limiting search direction. One can directly extend this safeguard update to the multiple-constraint scenario by doing gradient descent over the constraint that has the worst violation. Another remedy to reduce the chance of constraint violation is to do *constraint tightening* on the constraint cost threshold. Specifically, instead of $d_0$, one may pose the constraint based on $d_0 \cdot (1 - \delta)$, where $\delta \in (0, 1)$ is the factor of safety for providing additional buffer to constraint violation. Additional techniques in cost-shaping have been proposed in Achiam et al. (2017) to smooth out the sparse constraint costs. While these techniques can further ensure safety, construction of the cost-shaping term requires knowledge from the environment, which makes the safe PG algorithms more complicated.

# B  LAGRANGIAN APPROACH TO SAFE RL

There are a number of mild technical and notational assumptions which we will make throughout this section, so we state them here:

**Assumption 1** (Differentiability). *For any state-action pair $(x, a)$, $\pi_\theta(a|x)$ is continuously differentiable in $\theta$ and $\nabla_\theta \pi_\theta(a|x)$ is a Lipschitz function in $\theta$ for every $a \in \mathcal{A}$ and $x \in \mathcal{X}$.*

**Assumption 2** (Strict Feasibility). *There exists a transient policy $\pi_\theta(\cdot|x)$ such that $\mathcal{D}_{\pi_\theta}(x_0) < d_0$ in the constrained problem.*

**Assumption 3** (Step Sizes). *The step size schedules $\{\alpha_{3,k}\}$, $\{\alpha_{2,k}\}$, and $\{\alpha_{1,k}\}$ satisfy*

$$\sum_k \alpha_{1,k} = \sum_k \alpha_{2,k} = \sum_k \alpha_{3,k} = \infty, \tag{13}$$

$$\sum_k \alpha_{1,k}^2, \quad \sum_k \alpha_{2,k}^2, \quad \sum_k \alpha_{3,k}^2 < \infty, \tag{14}$$

$$\alpha_{1,k} = o(\alpha_{2,k}), \quad \zeta_2(i) = o(\alpha_{3,k}). \tag{15}$$

Assumption 1 imposes smoothness on the optimal policy. Assumption 2 guarantees the existence of a local saddle point in the Lagrangian analysis introduced in the next subsection. Assumption 3 refers to step sizes corresponding to policy updates that will be introduced for the algorithms in this paper, and indicates that the update corresponding to $\{\alpha_{3,k}\}$ is on the fastest time-scale, the updates corresponding to $\{\alpha_{2,k}\}$ is on the intermediate time-scale, and the update corresponding to $\{\alpha_{1,k}\}$ is on the slowest time-scale. As this assumption refer to user-defined parameters, they can always be chosen to be satisfied.

To solve the CMDP, we employ the Lagrangian relaxation procedure (Bertsekas, 1999) to convert it to the following unconstrained problem:

$$\max_{\lambda \geq 0} \min_\theta \left( L(\theta, \lambda) \triangleq \mathcal{C}_{\pi_\theta}(x_0) + \lambda \left( \mathcal{D}_{\pi_\theta}(x_0) - d_0 \right) \right), \tag{16}$$

where $\lambda$ is the Lagrange multiplier. Notice that $L(\theta, \lambda)$ is a linear function in $\lambda$. Then there exists a local saddle point $(\theta^*, \lambda^*)$ for the minimax optimization problem $\max_{\lambda \geq 0} \min_\theta L(\theta, \lambda)$, such that for some $r > 0$, $\forall \theta \in \mathbb{R}^\kappa \cap B_{\theta^*}(r)$ and $\forall \lambda \in [0, \lambda_{\max}]$, we have

$$L(\theta, \lambda^*) \geq L(\theta^*, \lambda^*) \geq L(\theta^*, \lambda), \tag{17}$$

where $B_{\theta^*}(r)$ is a hyper-dimensional ball centered at $\theta^*$ with radius $r > 0$.

In the following, we present a policy gradient (PG) algorithm and an actor-critic (AC) algorithm. While the PG algorithm updates its parameters after observing several trajectories, the AC algorithms are incremental and update their parameters at each time-step.

We now present a policy gradient algorithm to solve the optimization problem Equation 16. The idea of the algorithm is to descend in $\theta$ and ascend in $\lambda$ using the gradients of $L(\theta, \lambda)$ w.r.t. $\theta$ and $\lambda$, i.e.,

$$\nabla_\theta L(\theta, \lambda) = \nabla_\theta \left( \mathcal{C}_{\pi_\theta}(x_0) + \lambda \mathcal{D}_{\pi_\theta}(x_0) \right), \quad \nabla_\lambda L(\theta, \lambda) = \mathcal{D}_{\pi_\theta}(x_0) - d_0. \tag{18}$$

The unit of observation in this algorithm is a system trajectory generated by following policy $\pi_{\theta_k}$. At each iteration, the algorithm generates $N$ trajectories by following the current policy, uses them to estimate the gradients in Equation 18, and then uses these estimates to update the parameters $\theta, \lambda$.

Let $\xi = \{x_0, a_0, c_0, x_1, a_1, c_1, \ldots, x_{T-1}, a_{T-1}, c_{T-1}, x_T\}$ be a trajectory generated by following the policy $\theta$, where $x_T = x_{\text{Tar}}$ is the target state of the system and $T$ is the (random) stopping time. The cost, constraint cost, and probability of $\xi$ are defined as $\mathcal{C}(\xi) = \sum_{k=0}^{T-1} \gamma^k C(x_k, a_k)$, $\mathcal{D}(\xi) = \sum_{k=0}^{T-1} \gamma^k D(x_k, a_k)$, and $\mathbb{P}_\theta(\xi) = P_0(x_0) \prod_{k=0}^{T-1} \pi_\theta(a_k|x_k) P(x_{k+1}|x_k, a_k)$, respectively. Based on the definition of $\mathbb{P}_\theta(\xi)$, one obtains $\nabla_\theta \log \mathbb{P}_\theta(\xi) = \sum_{k=0}^{T-1} \nabla_\theta \log \pi_\theta(a_k|x_k)$.

Algorithm 2 contains the pseudo-code of our proposed policy gradient algorithm. What appears inside the parentheses on the right-hand-side of the update equations are the estimates of the gradients of $L(\theta, \lambda)$ w.r.t. $\theta, \lambda$ (estimates of the expressions in 18). Gradient estimates of the Lagrangian

function are given by

$$\nabla_\theta L(\theta, \lambda) = \sum_\xi \mathbb{P}_\theta(\xi) \cdot \nabla_\theta \log \mathbb{P}_\theta(\xi) \left( \mathcal{C}_{\pi_\theta}(\xi) + \lambda \mathcal{D}_{\pi_\theta}(\xi) \right),$$

$$\nabla_\lambda L(\theta, \lambda) = -d_0 + \sum_\xi \mathbb{P}_\theta(\xi) \cdot \mathcal{D}(\xi),$$

where the likelihood gradient is

$$\nabla_\theta \log \mathbb{P}_\theta(\xi) = \nabla_\theta \left\{ \sum_{k=0}^{T-1} \log P(x_{k+1}|x_k, a_k) + \log \pi_\theta(a_k|x_k) + \log \mathbf{1}\{x_0 = x^0\} \right\}$$

$$= \sum_{k=0}^{T-1} \nabla_\theta \log \pi_\theta(a_k|x_k) = \sum_{k=0}^{T-1} \frac{1}{\pi_\theta(a_k|x_k)} \nabla_\theta \pi_\theta(a_k|x_k).$$

In the algorithm, $\Gamma_\Lambda$ is a projection operator to $[0, \lambda_{\max}]$, i.e., $\Gamma_\Lambda(\lambda) = \arg\min_{\hat\lambda \in [0, \lambda_{\max}]} \|\lambda - \hat\lambda\|_2^2$, which ensures the convergence of the algorithm. Recall from Assumption 3 that the step-size schedules satisfy the standard conditions for stochastic approximation algorithms, and ensure that the policy parameter $\theta$ update is on the fast time-scale $\{\zeta_{2,i}\}$, and the Lagrange multiplier $\lambda$ update is on the slow time-scale $\{\zeta_{1,i}\}$. This results in a two time-scale stochastic approximation algorithm, which has shown to converge to a (local) saddle point of the objective function $L(\theta, \lambda)$. This convergence proof makes use of standard in many stochastic approximation theory, because in the limit when the step-size is sufficiently small, analyzing the convergence of PG is equivalent to analyzing the stability of an ordinary differential equation (ODE) w.r.t. its equilibrium point.

In policy gradient, the unit of observation is a system trajectory. This may result in high variance for the gradient estimates, especially when the length of the trajectories is long. To address this issue, we propose two actor-critic algorithms that use value function approximation in the gradient estimates and update the parameters incrementally (after each state-action transition). We present two actor-critic algorithms for optimizing Equation 16. These algorithms are still based on the above gradient estimates. Algorithm 3 contains the pseudo-code of these algorithms. The projection operator $\Gamma_\Lambda$ is necessary to ensure the convergence of the algorithms. Recall from Assumption 3 that the step-size schedules satisfy the standard conditions for stochastic approximation algorithms, and ensure that the critic update is on the fastest time-scale $\{\alpha_{3,k}\}$, the policy and $\theta$-update $\{\alpha_{2,k}\}$ is on the intermediate timescale, and finally the Lagrange multiplier update is on the slowest time-scale $\{\alpha_{1,k}\}$. This results in three time-scale stochastic approximation algorithms.

Using the policy gradient theorem from Sutton et al. (2000), one can show that

$$\nabla_\theta L(\theta, \lambda) = \nabla_\theta V_\theta(x_0) = \frac{1}{1-\gamma} \sum_{x,a} \mu_\theta(x, a|x_0) \nabla \log \pi_\theta(a|x) Q_\theta(x, a), \tag{19}$$

where $\mu_\theta$ is the discounted visiting distribution and $Q_\theta$ is the action-value function of policy $\theta$. We can show that $\frac{1}{1-\gamma} \nabla \log \pi_\theta(a_k|x_k) \cdot \delta_k$ is an unbiased estimate of $\nabla_\theta L(\theta, \lambda)$, where

$$\delta_k = \bar{C}_\lambda(x_k, a_k) + \gamma \widehat{V}(x_{k+1}) - \widehat{V}(x_k)$$

is the temporal-difference (TD) error, and $\widehat{V}$ is the value estimator of $V_\theta$.

Traditionally, for convergence guarantees in actor-critic algorithms, the critic uses linear approximation for the value function $V_\theta(x) \approx \phi^\top \psi(x) = \widetilde{V}_{\theta,\phi}(x)$, where the feature vector $\psi(\cdot)$ belongs to a low-dimensional space $\mathbb{R}^{\kappa_2}$. The linear approximation $\widetilde{V}_{\theta,\phi}$ belongs to a low-dimensional subspace $S_V = \{\Psi\phi | \phi \in \mathbb{R}^{\kappa_2}\}$, where $\Phi$ is a short-hand notation for the set of features, i.e., $\Psi(x) = \psi^\top(x)$. Recently with the advances of deep neural networks, it has become increasingly popular to model the critic with a deep neural network architecture, based on the objective function of minimizing the MSE of Bellman residual w.r.t. $V_\theta$ or $Q_\theta$ (Mnih et al., 2013).

## C    EXPERIMENTAL SETUP

Our experiments are performed on safety-augmented versions of standard MuJoCo domains (Todorov et al., 2012).

**HalfCheetah-Safe.** The agent is a the standard HalfCheetah (a 2-legged simulated robot rewarded for running at high speed) augmented with safety constraints. We choose the safety constraints to be defined on the speed limit. We constrain the speed to be less than 1, i.e., constraint cost is thus $\mathbf{1}[|v| > 1]$ . Episodes are of length 200. The constraint threshold is 50.

**Point Circle.** This environment is taken from (Achiam et al., 2017). The agent is a point mass (controlled via a pivot). The agent is initialized at $(0, 0)$ and rewarded for moving counter-clockwise along a circle of radius 15 according to the reward $\frac{-dx \cdot y + dy \cdot x}{1 + |\sqrt{x^2 + y^2} - 15|}$, for position $x, y$ and velocity $dx, dy$. The safety constraint is defined as the agent staying in a position satisfying $|x| \leq 2.5$. The constraint cost is thus $\mathbf{1}[|x| > 2.5]$. Episodes are of length 65. The constraint threshold is 7.

**Point Gather.** This environment is taken from (Achiam et al., 2017). The agent is a point mass (controlled via a pivot) and the environment includes randomly positioned apples (2 apples) and bombs (8 bombs). The agent given a reward of 10 for each apple collected and a penalty of $-10$ for each bomb. The safety constraint is defined as number of bombs collected during the episode. Episodes are of length 15. The constraint threshold is 2.

**Ant Gather.** This environment is is the same as Point Circle, only with an ant agent (quadrapedal simulated robot). Each episode is initialized with 8 apples and 8 bombs. The agent given a reward of 10 for each apple collected, a penalty reward of $-20$ for each bomb collected, and a penalty reward of $-20$ is incurred if the episode terminates prematurely (because the ant falls). Episodes are of length at most 500. The constraint threshold is 10 for DDPG algorithms and is 5 for PPO algorithms.

Figure 4 shows the visualization of the above domains used in our experiments.

| HalfCheetah-Safe | Point-Circle | Ant-Gather | Point-Gather |
| --- | --- | --- | --- |

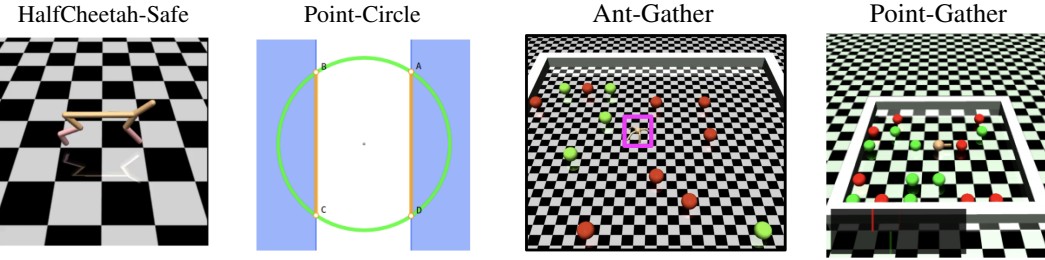

Figure 4: The Robot Locomotion Control Tasks

On top of the GAE parameter $\lambda$, in all numerical experiments and for each algorithm (SPPO, SDDPG, SPPO-modular, SDDPG-modular, CPO, Lagrangian, and the unconstrained PG counterparts), we systematically explored different parameter settings by doing grid-search over the following factors: (i) learning rates in the actor-critic algorithm, (ii) batch size, (iii) regularization parameters of the policy relative entropy term, (iv) with-or-without natural policy gradient updates, (v) with-or-without the emergency safeguard PG updates (see Appendix A.1 for more details). Although each algorithm might have a different parameter setting that leads to the optimal performance in training, the results reported here are the best ones for each algorithm, chosen by the same criteria (which is based on value of return plus certain degree of constraint satisfaction). To account for the variability during training, in each learning curve a 95% confidence interval is also computed over 10 separate random runs (under the same parameter setting).

### C.1    ADDITIONAL EXPERIMENTAL RESULTS

In this section, we include more experimental results of various PG algorithms on safe robot locomotion tasks.

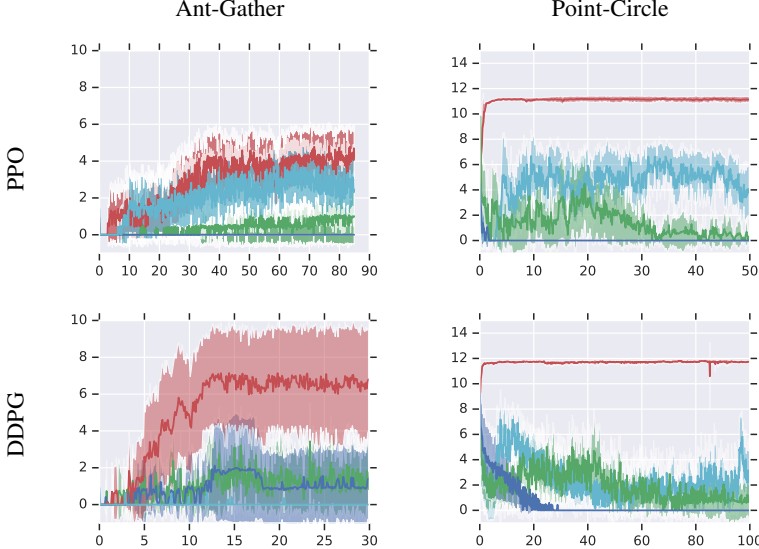

Figure 5: Constraint violations (logarithmic scale, i.e., $\log(1 + \mathrm{ccv})$) of various PG algorithms on safe robot locomotion tasks, with x-axis in thousands of episodes. We include runs from DDPG (red), DDPG-Lagrangian (magenta), SDDPG (blue), SDDPG-modular (green), PPO (red), PPO-Lagrangian (magenta), SPPO (blue), SPPO-modular (green) on Ant-Gather and Point-Circle.

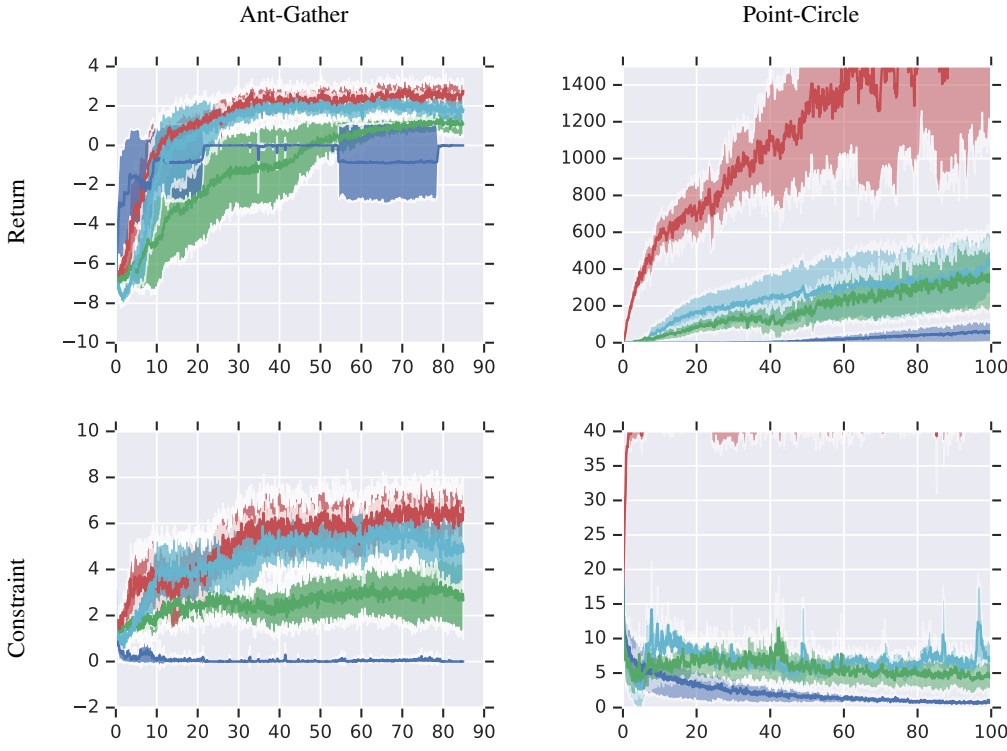

Figure 6: Results of various PG algorithms on safe robot locomotion tasks, with x-axis in thousands of episodes. We include runs from PPO (red), PPO-Lagrangian (magenta), SPPO (blue), SPPO-modular (green) on Ant-Gather and Point-Circle.

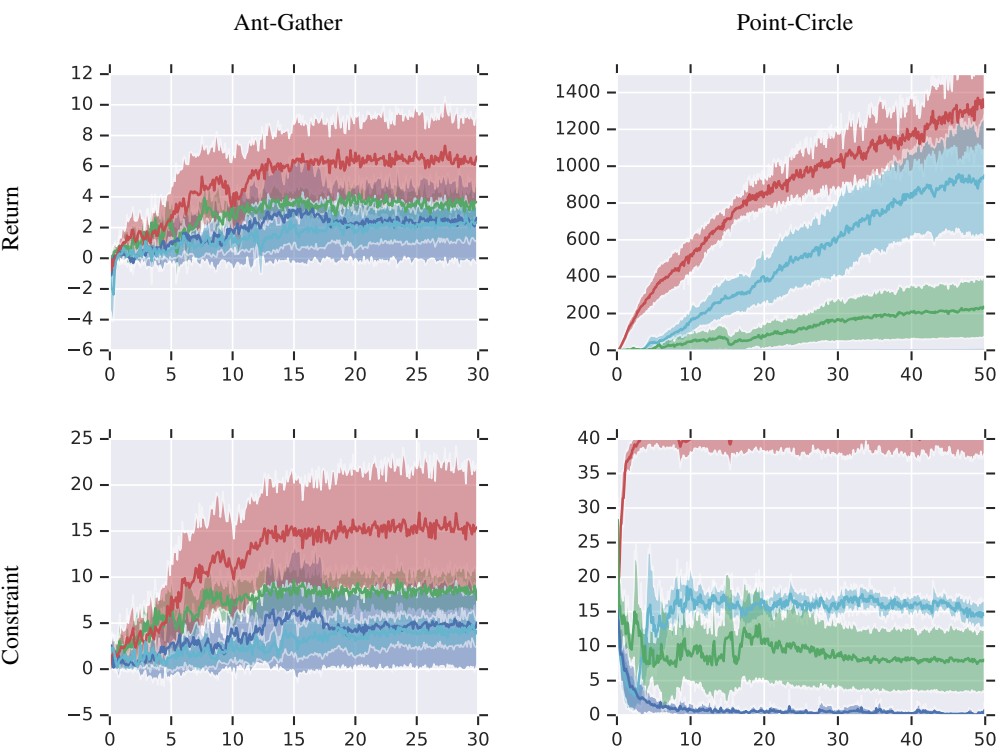

Figure 7: Results of various PG algorithms on safe robot locomotion tasks, with x-axis in thousands of episodes. We include runs from DDPG (red), DDPG-Lagrangian (magenta), SDDPG (blue), SDDPG-modular (green) on Ant-Gather and Point-Circle.

