# OpenReview forum: "Lyapunov-based Safe Policy Optimization"
_ICLR.cc/2019/Conference_

### Official Review · AnonReviewer3 · 2018-11-02

**Rating:** 5
**Confidence:** 3

**Review:**

￼
To my understanding, this paper builds on prior work from Chow et al. to apply Lyapunov-based safe optimization to the policy-gradient setting. This seems is similar to work by Achiam 2017. While this work seems like an interesting framework for encompassing several classes of constrained policy optimization settings in the Lyapunov-based setting, I have some concerns about the evaluation methodology.

It is claimed that the paper compares against “two baselines, CPO and the Lagrangian method, on several robot locomotion tasks, in which the agent must satisfy certain safety constraints while minimizing its expected cumulative cost.” Then it is stated in the experimental section “However since backtracking line-search in TRPO can be computationally expensive, and it may lead to conservative policy updates, without loss of generality we adopt the original construction of CPO to create a PPO counterpart of CPO (which coincides with SPPO) and use that as our baseline.” This seems directly to contrast to the earlier statement which states that it is unclear how to modify the CPO methodology to other RL algorithms. Moreover, is this really a fair comparison? The original method has been modified to form a new baseline and I’m not sure that it is “without loss of generality”.

Also, it is unclear whether the results can be accepted at face value. Are these averaged across several random seeds and trials? Will performance hold across them? What would be the variance? Recent work has shown that taking 1 run especially in MuJoCo environments doesn’t necessarily provide statistically significant values. In fact the original CPO paper shows the standard deviations across several random seeds and compares directly against an earlier work in this way (PDO). Moreover, it is unclear why CPO was not directly compared against and neither was the "equivalent" baseline not compared on similar environments as in the original CPO paper.

Comments:

Figure 3 is difficult to parse, the ends of the graphs are cut off. Maybe putting the y axis into log format would help with readability here or having the metrics be in a table.

---

> ### Public Comment · (anonymous) · 2018-11-26
> **Thank you, our response regarding comparison with CPO**
>
> We thank the reviewer for useful comments. We are glad that the reviewer found the proposed framework interesting as a novel generalization of several existing safe RL algorithms. Below is our response to the reviewer’s comments. We hope that with the additional results, the reviewer will find the evaluation of the paper stronger and more satisfactory.
>
>
> “This seems directly to contrast to the earlier statement which states that it is unclear how to modify the CPO methodology to other RL algorithms”
> The reviewer is absolutely right. We realize that the current wording is not quite accurate and need to be revised. The main message we aim to deliver is that while one can easily transfer ideas from CPO (based on TRPO) to PPO, to the best of our knowledge there are no direct ways to further apply this idea to more general policy gradient algorithms, such as DDPG, in order to use them to solve CMDPs.
>
> We hereby updated the corresponding part of the introduction to:
> “While it is straightforward to adopt this methodology to PPO for constrained optimization (this is exactly how we derive the SPPO algorithm in our paper), it is unclear how to combine it with algorithms that do not belong to the family of proximal PG algorithms (i.e., PG algorithms that are regularized with relative entropy), such as DDPG.”
>
> “Fair comparison with CPO and without loss of generality?”
> Below is what we think about our empirical evaluation and comparison with CPO:
>
> The essence of CPO, which is to add a first order constraint in proximal policy optimization, is preserved in the SPPO algorithm. In fact in Section 4.1, we show that it is a special case of the Lyapunov-based approach. In fact, besides using the relative entropy penalty update schedules in PPO, we also ran experiments based on the relative entropy penalty term suggested by TRPO (as shown in Appendix C of the TRPO paper). We treated this choice as one of the hyper-parameters in the algorithm (see the response to Reviewer 2 about details on systematic comparisons) and reported the best ones as performance of SPPO. The only difference between CPO and SPPO is that SPPO does not perform backtracking line-search.
> Unfortunately, the computational complexity of PG is significantly increased with line-search and the corresponding CPO algorithm is not back-propagatable. This is why we decided to test the safety algorithms based on the popular PPO algorithm (which belongs to the family of proximal PG algorithms) instead of TRPO, without losing the essence of the ideas behind CPO.
> We did try to reimplement the original CPO algorithm, but we did not obtain the results reported in the CPO paper. We also ran into trouble understanding which part of the original CPO implementation in rllab contributes to the difference, and thus, found their comparisons difficult with the safe RL algorithms, which are not implemented in rllab.
> We think that the CPO modification from TRPO-based to PPO-based algorithms is indeed needed in our experiments, because otherwise their comparison with safe DDPG-based algorithms and unconstrained PG algorithms with safety layer (see Section 4.2), which do not perform backtracking line-search, may be unfair.
>
>
> While we think our comparison with CPO is valid, we agree with the reviewer that the claim of “without loss of generality” is too strong. Given the current results, we view the Lyapunov safe RL algorithms as an alternative to CPO and will make sure that we deliver this message in our paper. We summarized the above points and modified the sentence of justifying the switch from TRPO to PPO as follows:
> “Instead of the original CPO algorithm, we use its PPO alternative (which coincides with the SPPO algorithm derived in Section 4.1) as the safe RL baseline for comparison. SPPO preserves the essence of CPO by adding a 1st order constraint to the proximal policy optimization. The main difference between CPO and SPPO is that the latter does not perform backtracking line-search. The decision to compare with SPPO instead of CPO is 1) to avoid the additional computational complexity of line-search in TRPO, while maintaining the performance of PG using the popular PPO algorithm, 2) to have a back-propagatable version of CPO, and 3) to have a fair comparison with other back-propagatable safe RL algorithms, such as the DDPG and safety layer counterparts.“
>
> "Experiments, Figure 3, 10 Random seeds, statistical significance, and variance”
> Thank you for the suggestions to improve the readability of results. We updated all figures in the paper, added DDPG for PointCircle, added confidence intervals (over 10 random seeds) to the learning curves, and did log-transformation to the y-label of Figure 3. Although we haven't run all the experiments from the CPO paper, we did increase the difficulty of some tasks (e.g., AntGather) and try both DDPG and PPO safe RL algorithms in each domain for comprehensive comparisons. The results still deliver similar message as in the old revision.

---

> > ### Comment · AnonReviewer3 · 2018-11-27
> > **Appreciate the updates**
> >
> > I appreciate the updates to the work. While some of my concerns remain regarding the comparison with CPO, I think the recent revision is improved and I'll be updating my original rating to reflect this.

---

> > > ### Public Comment · (anonymous) · 2018-11-28
> > > **Thank you. Please find our additional remarks about CPO**
> > >
> > > We thank the reviewer for going over our response and adjusting her/his score accordingly. We are happy that we managed to address some of her/his concerns.
> > >
> > > Regarding comparison with CPO: Unfortunately, the current version of CPO on github is built in rllab, and no implementation of this algorithm is available outside this setting. As a result, a fair comparison with CPO requires implementing our algorithms in rllab, which is rather difficult and time consuming. This is why we used SPPO (the PPO alternative of CPO) as a replacement for CPO in our experiments. Finally, we would like emphasize that the message of the paper is NOT to advocate our Lyapunov-based algorithms as a replacement for CPO, but rather as an alternative to CPO for solving CMDPs with continuous actions, when safety (not violating the constraints) is critical even during training.

---

### Official Review · AnonReviewer1 · 2018-11-05
**Incremental, but quite solid**

**Rating:** 8
**Confidence:** 3

**Review:**

In this paper, authors propose safe policy optimization algorithms based on the Lyapunov approach to constrained Markov decision processes.
The paper is very well written (a few typos here and there, please revise) and structured, and, to the best of my knowledge, it is technically sound and very detailed.
It provides incremental advances, mostly from Chow et al., 2018.
It fairly accounts for recent literature in the field.
Experimental settings and results are fairly convincing.

Minor issues:
Authors should not use not-previously-described acronyms (as in the abstract: DDPG, PPO, PG, CPO)

---

> ### Public Comment · (anonymous) · 2018-11-20
> **Thank you**
>
> We thank the reviewer for the useful comments.  We are glad that the reviewer found the paper interesting and the experimental evaluation convincing.  We appreciate the comments regarding clarity of the writing - we will fix the typos.  We have also updated the paper to appropriately introduce the acronyms (DDPG, PPO, PG, CPO). Please see the updated version of the paper.
>
> “Incremental advances”
> Regarding the contributions of this paper, the main objective here is to present a general and unified method for deriving safe RL algorithms in problems with continuous actions. While we base our approach on ideas from Lyapunov theory (previously applied to discrete-control problems in Chow et al., 2018), we believe our theoretical and empirical  results to be significant for several reasons:
> 1) The focus of Chow et al, 2018 is to derive value-based safe RL algorithms which are more suitable for solving problems with discrete action spaces.  In general, it is unclear how to apply the same techniques to continuous control problems, which are more common in robotics applications. This is the main issue we address in our paper.
> 2) Compared to CPO algorithm (Achiam et al 2017), our work can be applied to the off-policy setting. Therefore, our Lyapunov-based policy gradient algorithms are more data-efficient, as one can utilize data from replay buffer. Furthermore, from the derivations in Section 4.1,  it would be possible to view CPO as a special case of Lyapunov-based PG, which is an interesting result by itself.
> 3) Compared to Dalal et al 2018, which (to our knowledge) is the first work to propose the idea of a safety layer, our work is more general. While they focus on constraints that can only be expressed locally, we solve the more general CMDP problem (with trajectory-based constraints). Our paper provides an elegant recipe (through the use of Lyapunov functions) to adopt the safety layer concept to solve safe RL problems with general trajectory-based constraints using direct backpropagation.

---

### Official Review · AnonReviewer2 · 2018-11-06
**entirely reasonable paper, but novelty is unclear, empirical verification incomplete**

**Rating:** 6
**Confidence:** 2

**Review:**

In this paper, authors compare different ways to enforce stability constraints on trajectories in dynamic RL problems. It builds on a recent approach by Achiam et al on Constrained Policy Optimization (oft- mentioned "CPO") and an accepted NIPS paper by Chow which introduces Lyapunov constraints as an alternative method. While this approach is reasonable indeed, the novelty of the approach is questionable, not only in light of recent papers but older literature: inference of Markov Decision Processes under constraints is referred to and has been known a long time. Furthermore, the actual tasks chosen are quite simple and do not present complex instabilities. Also, actually creating a Lyapunov function and weighing the relative magnitude of its second derivative (steep/shallow) is not trivial and must influence the behavior of the optimizer. Also worth mentioning that complex nonlinearities might imply that instabilities in the observed dynamics are not seen and learned unless the space exploration is conservative. That is, comparison of CPO and Lagrangian constraint based RL with Lyapunov based method proposed depends on a lot of factors (such as those just mentioned) that are not systematically explored by the paper.

---

> ### Public Comment · (anonymous) · 2018-11-21
> **Thank you, please find our response to your questions below**
>
>
> We thank the reviewer for the useful comments. Below is our response to the reviewer’s main comments.
>
> “Questionable Contributions”
> Due to the space limit, please see the list of our contributions in our response to Reviewer 1.
>
> “Comparison with inference of Markov Decision Processes under constraints”
> For solving CMDPs, Chow et al, 2018 provided a good literature survey on existing methods. Specifically the Lyapunov-based safe RL algorithms are hinged on the ``primal method’’, which aims to learn the value and Lyapunov functions. On the other hand, there is also the ``dual method’’ that learns the occupation measure. However, this algorithm requires the knowledge of the dynamics and the LP-based algorithm is limited to solving CMDP problems with finite state and action spaces, which is different than what we are interested in here.
>
> Regarding the inference perspective of MDPs, we looked into some of the standard formulations. While this approach may resemble the spirit of learning occupation measures, it appears that the MDP problems studied in the literature are mostly unconstrained (for example, see https://ipvs.informatik.uni-stuttgart.de/mlr/marc/publications/06-toussaint-ICML.pdf). Furthermore, they usually derive stochastic optimal control algorithms by assuming Gaussian noise and linear dynamics. It would helpful if the reviewer mention the references (s)he has in mind that we can look into and cite them.
>
> “Creating a Lyapunov function”
> We agree with the reviewer that the choice of the Lyapunov function is quite important. This is why we learn the Lyapunov function in our algorithms and update it as the algorithms progress. The reviewer’s comment regarding the 2nd derivative of the Lyapunov function is unclear to us. Similar to the standard policy gradient algorithms (DDPG, PPO), we only use the first order information of the objective and Lyapunov functions in all the safe RL algorithms used in the paper. To make sure that the first order taylor series expansions are good enough approximations, similar to the trust region method (such as TRPO), we restrict the local policy update using a quadratic constraint that is constructed via second-order gradient of the relative entropy term. But this has nothing to do with the Lyapunov and Q functions.
>
> “Complex nonlinearities and instability”
> In this work, we mainly focus on the derivation of model-free RL algorithms with safety guarantees (w.r.t. CMDP constraints), especially when the action space is continuous. To guarantee constraint satisfaction and safety during training (which may restrict the exploration to be conservative), we leverage the recent results of Lyapunov functions from Chow et al (2018) and extend it to policy gradient methods. Although Lyapunov functions are used here, it is not for stability, it is for bounding the constraint performance of an MDP. A similar work is https://pdfs.semanticscholar.org/9b24/d6a26526d9a02168432988060ba6721ff926.pdf. While stability of nonlinear dynamics is an important safety criterion, it is NOT the focus of this paper. That being said, we believe RL with stability guarantees is an interesting future direction, especially in model-based RL.
>
> “the actual tasks chosen are quite simple”
> We agree that the chosen tasks are simple control problems, but indeed they are among the standard RL benchmarks and have been used by similar papers, including CPO (with which we would like to make comparisons). While these tasks (besides HalfCheetah) might not have complex instabilities in the dynamics, our major focus in the experiments is to evaluate the proposed safety algorithms in terms of the return maximization and constraint satisfaction during training. To handle more complicated/realistic control tasks, we speculate model-based RL algorithms might be more suitable. We leave this important research direction for future work.
>
> “Comparison are not systematically explored by the paper.”
> In all numerical experiments and for each algorithm (SPPO, SDDPG, SPPO-modular, SDDPG-modular, CPO, Lagrangian, and the unconstrained PG counterparts), we systematically explored different settings by doing grid-search over the following factors: (i) learning rates in the actor-critic algorithm, (ii) batch size, (iii) regularization parameters of the policy relative entropy term, (iv) with-or-without natural policy gradient updates, (v) with-or-without the emergency safeguard PG updates (see Appendix A for more details). Although each algorithm might have a different parameter setting that leads to the optimal performance in training, the results reported in the paper are the best ones for each algorithm, chosen by the same criteria (which is based on value of return + degree of constraint satisfaction). We also add this description in Appendix C for clarification. Please also check the updated numerical results in the revised paper.

---

### Official Review · AnonReviewer4 · 2018-11-27

**Rating:** 6
**Confidence:** 2

**Review:**

The paper generalized the approach for safe RL by Chow et al, leveraging Lyapunov functions to solve constraint MDPs and integrating this approach into policy optimization algorithms that can work with continuous action spaces.
This work derives two classes of safe RL methods based on Lyapunov function, one based on constrained parameter policy optimization (called theta-projection), the other based on a safety layer. These methods are evaluated against Lagrangian-based counter-parts on 3 simple domains.

The proposed Lyapunov-function based approach for policy optimization is certainly appealing and the derived algorithms make sense to me. This paper provides a solid contribution to the safe RL literature though there are two main reasons that dampen the excitement about the presented results:
First, the contribution is solid but seems to be of somewhat incremental nature to me, combining existing recent techniques by Chow et al (2018) [Lyapunov-function based RL in CMDP], Achiam et al (2017) [CPO with PPO is identical to SPPO] and Dalal et al (2018) [safety layer].
Second, the experimental results do not seem to show a drastic benefit over the Lagrangian baselines. It is for example unclear to me whether the jiggling of the Lagrange approach around the threshold is a problem is practice. Further, it seems that PPO-Lagrangian can achieve much higher performance in the Point and Gather task while approximately staying under the constraint threshold of 2. Also, as far as I understand, the experiments are based on extensive grid search over hyper-parameters including learning rates and regularization parameters. However, it is not clear why the initialization of the Lagrange multiplier for the Lagrangian baselines was chosen fixed. I would be curious to see the results w.r.t. the best initial multiplier or a comparison without hyper-parameter grid search at all.

This is a light review based on a brief read.

Minor note:
In the figures: where is the magenta line? I assume magenta labels refer to teal lines?

---

> ### Public Comment · (anonymous) · 2018-11-28
> **Thank you, please find our response to your questions below**
>
> We thank the reviewer for useful comments. Below is our response to the reviewer’s comments.
>
> "contribution is incremental"
> It is true that the paper is built based on the Lyapunov-function-based approach to CMDPs from Chow et al., 2018, and borrows ideas from Dalal et al., 2018 and CPO (Achiam et al., 2017). However, we believe that extending the setting of Chow et al. 2018 to continuous actions is not straightforward, our improvement over the safety layer idea of Dalal et al. 2018 is significant, and finally we have clear differences with CPO. Below is a detailed comparison of our work with each of these papers. We will make the comparisons and our contributions more clear in the final version of the paper.
>
> 1) The Lyapunov-function-based algorithms of Chow et al., 2018 were all value-function-based (approximate policy and value iteration), and thus, could not easily handle continuous action CMDPs. In order to handle continuous actions, we had to develop policy gradient and actor-critic type algorithms based on the Lyapunov formulation, which is not a straightforward extension of the results of Chow et al., 2018.
> 2) Dalal et al., 2018 proposed the idea of using a safety layer for constraints, but their results were restricted to constraints that can be expressed locally. We show how this idea can be used to solve more general CMDPs with trajectory-based constraints. This provides an elegant recipe (through the use of Lyapunov functions) to adopt the safety layer concept and solve CMDPs with direct back-propagation.
> 3) Compared to CPO, our Lyapunov-based policy gradient algorithms can be used in the off-policy setting, which makes them more data-efficient, as they can utilize the data from the replay buffer. Moreover, we show in Section 4.1 that CPO can be viewed as a special case of the Lyapunov-based approach. From an application standpoint, since all the algorithms proposed in this paper are back-propagatable, it might be more efficient to implement them in TensorFlow and PyTorch, than the original (TRPO-based) CPO, which is not back-propagatable.
>
> “Experiments and comparisons with Lagrangian approach”
> In our experiments, we compare our two safe RL algorithms, one derived from constrained optimization and one from the safety layer idea, with the unconstrained and Lagrangian baselines in four problems: PointGather, AntGather, PointCircle, and HalfCheetahSafe. We perform these experiments with both off-policy (DDPG version) and on-policy (PPO version) versions of the algorithms.
>
> In PointCircle DDPG, although the Lagrangian algorithm significantly outperforms the safe RL algorithms in terms of return, it violates the constraint more often. The only experiment in which Lagrangian performs similarly to the safe algorithms in terms of both return and constraint violation is PointCircle PPO. In other experiments, either 1) the policy learned by Lagrangian has a significantly lower performance than that learned by one of the safe algorithms (see HalfCheetahSafe DDPG, PointGather DDPG, AntGather DDPG), or 2) the Lagrangian method violates the constraint during training, while the safe algorithms do not (see HalfCheetahSafe PPO, PointGather PPO, AntGather PPO). We will make these comparisons more clear in the paper.
>
> We admit that the differences are not quite significant in the standard benchmarks used in our experiments (similar to the experiments in most papers on this topic). It would interesting to evaluate all these algorithms in real problems, for example in robotics, but this is a separate contribution that we leave for future work.
>
> “Why is jiggling a problem in practice”
> Jiggling around the threshold means that the algorithm generates policies that violate the constraints during training. This might be ok in certain applications, but there are problems in which it would be critical to violate the constraints even during training. This is one of the major problems of using Lagrangian algorithms to solve CMDPs. As it was mentioned in the introduction, similar to CPO and Chow et al., 2018, one of our goals is to develop CMDP algorithms that do not violate the constraints during training (or to reduce the violation as much as possible, as achieving this goal is difficult, in particular with function approximations in complex domains). We will make this more clear in the paper.
>
> “Grid-search over initialization Lagrange multiplier”
> The reviewer is right, we did not do a grid-search over the initial Lagrange multiplier. The main reason is that we tried a few values and used heuristics to balance the learning progress of reward and constraint reward, but we did not observe a significant difference in the performance and constraint violation, once learning was stabilized. However, we totally agree that a more systematic search over this parameter would result in a better comparison with the Lagrangian algorithms.
>
> “Magenta versus Teal”
> Thanks for pointing this out. We will correct this in the paper.

---

### Meta-Review · Area_Chair1 · 2018-12-14
**Important topic, limited novelty**

**Confidence:** 3
**Recommendation:** Reject

**Metareview:**

This is an interesting direction but multiple reviewers had concerns about the amount of novelty in the current work, and given the strong pool of other papers, this didn't quite reach the threshold.